# The Epigenetic Mechanisms Underlying Thermomorphogenesis and Heat Stress Responses in *Arabidopsis*

**DOI:** 10.3390/plants10112439

**Published:** 2021-11-12

**Authors:** Anna Zioutopoulou, Eirini Patitaki, Tianyuan Xu, Eirini Kaiserli

**Affiliations:** Institute of Molecular, Cell and Systems Biology, College of Medical, Veterinary and Life Sciences, University of Glasgow, Glasgow G12 8QQ, UK; 2346572z@student.gla.ac.uk (A.Z.); 2414878p@student.gla.ac.uk (E.P.); 2672494x@student.gla.ac.uk (T.X.)

**Keywords:** epigenetics, thermomorphogenesis, heat stress, *Arabidopsis*, environmental adaptation

## Abstract

Integration of temperature cues is crucial for plant survival and adaptation. Global warming is a prevalent issue, especially in modern agriculture, since the global rise in average temperature is expected to impact crop productivity worldwide. Hence, better understanding of the mechanisms by which plants respond to warmer temperatures is very important. This review focuses on the epigenetic mechanisms implicated in plant responses to high temperature and distinguishes the different epigenetic events that occur at warmer average temperatures, leading to thermomorphogenic responses, or subjected to extreme warm temperatures, leading to heat stress.

## 1. Introduction

Climate change combined with a constant increase in global temperature levels are current phenomena that intensify annually (NASA Earth Observatory). An increase in the average levels of global temperature is expected to have a considerable impact on food security by limiting crop yield and productivity [1,2]. Thus, it is important to understand and better decipher the molecular and physiological mechanisms that plants employ to adapt in response to warmer temperatures. Studies on the model plant *Arabidopsis thaliana* have been key for exploring warm temperature and heat response mechanisms. The potential applications of novel findings can be translated as a means of increasing crop yield via precision agriculture and gene editing technologies. *Arabidopsis*
*thaliana* is universally used as a model plant organism, due to its many advantages (well characterized small genome, quick growth, production of many generations and seeds, very efficient transformation methods). Additionally, genes that are isolated from *Arabidopsis thaliana* can be used to identify homologs in many commercial crops, thus establishing research in *Arabidopsis thaliana* is pivotal for further understanding plant crop improvement.

High temperature response mechanisms in plants can be categorized in two classes: (a) heat stress responses or (b) thermomorphogenic responses, depending on the temperature range to which plants are subjected. Thermomorphogenesis is defined as a collection of physiological and architectural changes such as stem extension of both the hypocotyls and petioles of the plant, early flowering initiation and leaf hyponasty, leading to heat dissipation [3,4]. Thermomorphogenesis can be categorized into three different groups [3]. The first two groups include responses initiated by prolonged periods of cold temperature (vernalization process) [5] or fluctuating temperatures [6,7]. The third group includes the general responses and effects of prolonged mildly warmer average temperature (22–32 °C), on the overall development and growth of the plant [3], and, will be discussed in this review along with heat stress responses. To date there are a number of thermo-sensing proteins and mechanisms which have been identified in *Arabidopsis*, such as the red/far-red light receptor and thermosensor phytochrome B (phyB) [8,9], the evening complex components EARLY FLOWERING 3 (ELF3) [10], alternative splicing of the PHYTOCHROME INTERACTING FACTOR 7 (PIF7) [11], and the exclusion of the histone variant H2A.Z [12], all of which regulate major developmental and architectural responses to warmer temperature. 

Heat stress responses in plants occur under extreme warm temperature conditions (exceeding 40 °C) [13]. Heat stress responses are orchestrated by the HSF (Heat Shock Transcription Factor) protein family, which induce the expression of heat stress proteins (HSPs), to promote protein homeostasis [14,15,16]. HSFs have crucial roles during the initial response to high temperature, the recovery stage and memory acquisition [17]. HSFA2 (HEAT SHOCK TRANSCRIPTION FACTOR A2) for example, is the most important component in the regulation of transcriptional memory after heat stress [17].

Thermomorphogenesis and heat stress responses are regulated by changes in the epigenetic status of histones and DNA, which leads to the control of gene expression. This occurs through the regulation of DNA transcription at the post-translational level, initiated by chemical modifications to histone proteins [18]. Epigenetic modifications include DNA and histone methylation [19], histone acetylation [20], phosphorylation [21,22], ubiquitination [23] and SUMOylation [24]. Additional modifications may also include regulation of non-coding RNAs (ncRNAs), as well as RNA-mediated DNA methylation (RdDM) and transcriptional gene silencing [25].

Acetylation is probably the most established and best-studied histone modification that has an effect on transcriptional regulation [20]. Most commonly the role of acetylation is to assign a negative charge to lysine residues on the N-terminal histone tails found at the periphery of the nucleosome [20]. Histone acetylation predominantly results in a more relaxed chromatin structure, which enhances binding of the transcriptional machinery and consequently leads to an upregulation of gene expression [20]. Acetylation is controlled by histone acetyltransferases (HAT) and histone deacetylases (HDAC) [20].

Histone methylation refers to the addition of methyl groups to the lysine or arginine residues of histones H3 and H4 [19]. Histone methylation, especially on lysine residues, can either promote or repress transcriptional activation depending on the nature of the modification [19]. On top of that, the possibility of di- or tri- methylation further adds to the functional diversity of this modification [19]. For example H3K27me3 is a common repressive signal at promoter regions of genes that control developmental processes [19]. 

Histone phosphorylation facilitates the deposition and interaction with other histone modifications and leads to the initiation of a variety of downstream responses including DNA damage repair and the regulation of chromatin architecture [21,22]. Histone ubiquitination is also involved in DNA damage responses [23], whereas chromatin-associated SUMOylation in plants plays a role in regulating development and various stress responses [24]. There are many potential applications of modulating chromatin-associated SUMOylation as a tool for crop improvement under stress-inducing conditions, thus making the study of molecular mechanisms involved in SUMO-dependent stress response regulation an important research topic [24,26].

All of the above epigenetic mechanisms play important roles in plant heat stress responses and/or thermomorphogenesis. The current review discusses recent studies depicting novel findings on the epigenetic regulation of high temperature-mediated plant responses.

## 2. Epigenetic Regulation of Thermo-Regulated Responses in *Arabidopsis*

### 2.1. Chromatin Regulation in Response to Warm Ambient Temperature

#### 2.1.1. Chromatin Remodeling and Histone Modifications

Chromatin remodeling and/or re-organization constitutes an important factor of warm temperature sensing and subsequent signaling [27]. Some important events which occur upon exposure of plants to warmer temperatures, include the exclusion of the histone variant H2A.Z from +1 position nucleosomes on the promoter of the growth-promoting PHYTOCHROME INTERACTING FACTOR 4 (*PIF4*) [12], as well as the auxin biosynthesis regulator and hypocotyl elongation promoter *YUCCA8* [4,28]. Exclusion of H2A.Z enhances chromatin accessibility and permits PIF4 to bind the G-box promoter element and induce the expression of auxin-responsive and hypocotyl elongation genes, such as *IAA29* (*INDOLE ACETIC ACID-INDUCED PROTEIN 29*) [29] (Figure 1). The upregulation of the expression of genes such as *PIF4* and *YUCCA8* that promote hypocotyl elongation in response to elevated temperature is also facilitated by POWERDRESS (PWR), HISTONE DEACETYLASE 9 (HDA9) (Figure 1) and HEAT SHOCK PROTEIN A1 [12,27,30,31]. A recent study has reported that HDA9, which is accumulated under warmer temperatures, is also needed for the induction of *YUCCA8* expression [2] (Figure 1). More specifically, both *YUCCA8* expression and auxin accumulation are triggered by H3K9K14 histone deacetylation mediated by HDA9 [2]. When HDA9 is not active, thermomorphogenesis is subsequently attenuated [2].

PWR is also involved in the initiation of thermomorphogenesis and the transcriptional regulation of *PIF4* and *YUCCA8*, via the facilitation of histone H3K9K14 deacetylation [12,32]. PWR physically interacts with HDA9 leading to histone deacetylation of certain loci genome-wide [27] (Figure 1). Specifically, *PWR* acts upstream of *PIF4* as a chromatin remodeler (Figure 1), and, it has been shown that the absence of *PWR* results in excess acetylation of the *PIF4* locus with a subsequent reduction in *PIF4* gene expression [27]. Therefore, histone H3K9K14 deacetylation, appears to be required for the nucleosome eviction of H2A.Z at the +1 nucleosome of locus *PIF4* [27]. Moreover, transcriptome analysis and statistical association studies, indicated a link between H2A.Z nucleosome dynamics and histone deacetylation mediated by *PWR* [27]. However, further analysis is required to examine the exact relationship between H2A.Z exclusion and H3K9K14 deacetylation. 

In addition to a constant or prolonged increase in temperature, a common phenomenon to which plants are subjected to when growing under natural conditions is recurring temperature fluctuations [3,33,34]. A very recent study revealed that a family of demethylases called JUMONJI (JMJ), is involved in mediating the epigenetic memory in response to high temperature in *Arabidopsis* [35]. In particular, JMJ proteins are necessary for removing H3K27me3 marks from genes responsible for maintaining warm temperature and heat memory [35]. Epigenetic memory or epi-priming can lead to a rapid and efficient response to a particular environmental stimulus, such as elevated temperature compared to the initial exposure [36].

#### 2.1.2. RNA Editing and Regulation in Response to High Temperature

Regulatory RNA thermo-switches in plants are a novel field of research. A recent study has revealed that PIF7 plays a role in the activation of thermomorphogenic responses [11]. Specifically, this activation occurs through the formation of an RNA hairpin at the 5′ end untranslated region of *PIF7*, which leads to an increase in protein synthesis and accumulation of PIF7 under warm day temperatures [11]. Warmer day temperatures lead to the relaxation of secondary structure of the RNA hairpin that becomes partially unfolded [11]. The more relaxed and unfolded conformation of the hairpin facilitates translation and leads to higher PIF7 protein levels [11]. PIF7 protein then induces the transcription of *YUCCA8* and *IAA* genes (*IAA19* and *IAA29*), resulting in an upregulation of the protein levels synthesized by these genes [11]. 

### 2.2. Heat Stress Mediated Chromatin Events

#### 2.2.1. Heat-Mediated Chromatin Remodeling

Chromatin dynamics play a pivotal role in the regulation of global- and locus-specific gene expression during heat stress responses. FGT1 (FORGETTER 1), which encodes for a PHD (plant homeodomain) finger protein, physically interacts with chromatin remodelers around the region which surrounds the transcription start sites (TSS) of heat-responsive genes to promote heat-induced memory [37]. In particular, FGT1 functions in synergy with BRAHMA (BRM), CHROMATIN-REMODELLING PROTEINS 11 and 17 (CHR11 and CHR17) to modulate nucleosome dynamics at the gene loci of HSP22, HSP18.2, HSP21 and HSA32 (HEAT-STRESS-ASSOCIATED 32), and induce their expression after the initial heat stress [37] (Figure 2). Furthermore, when *Arabidopsis* plants were subjected to heat stress, chromatin-associated SUMO signals increased around TSS regions that are enriched in active histone marks [26]. Intriguingly, chromatin SUMOylation facilitates the transcriptional reprogramming from plant growth to stress responses, as indicated by the overexpression of SUMO-associated heat-responsive genes and the downregulation of growth-promoting genes during heat stress [26] (Figure 2). 

#### 2.2.2. Nuclear Re-Organization

Heat stress (HS) has been reported to influence the architecture of the plant nucleus. Dense fibrillar component (DFC) diffusion is a nuclear process which was observed in the nucleolus under HS treatment and has been shown to be reversible during HS recovery [38,39]. Heat-regulated DFC diffusion has been associated with potential changes in liquid-liquid phase separation (LLPS) of intrinsically disordered regions (IDRs) in *Arabidopsis* FIBRILLARIN 2 (FIB2) and NUCLEOLIN LIKE 1 (NUC1) proteins [38]. Oppositely, the nucleolus-associated domains (NADs) are stable and not influenced by HS and cell senescence [38]. 

Prolonged exposure to high temperatures also causes chromocenter de-condensation and dispersion of heterochromatin, which is directly linked to TE (Transposable Element) activation [40,41,42,43]. HIT4 (HEAT-INTOLERANT 4), a pivotal regulator of chromatin re-organization in response to heat stress, facilitates nucleosome dispersion leading to the release of transcriptional gene silencing (TGS) [42]. A recent study demonstrated that in *Arabidopsis*, heat stress markedly affects chromatin dynamics by triggering global re-organization. In addition, local chromatin interactions are enhanced whilst distant intrachromosomal interactions diminish, with the latter being associated with reduced chromatin compartmentalization and TE activation [41].

#### 2.2.3. Histone Modifications 

Histone modifications, such as histone methylation and acetylation govern transcriptional regulation and epigenetic memory in response to heat stress. Interestingly, chromatin immunoprecipitation analysis demonstrated a significant depletion in global deposition of H3K9me2 and H3K4me3 repressing marks in *Arabidopsis* plants subjected to prolonged heat stress treatments [40]. As mentioned in the introduction, an important HSF factor, HSFA2, forms heterodimers with HSFA3 and interacts with additional HSFs to recruit H3K4 histone methyltransferases and mediate histone H3K4 hypermethylation to maintain transcriptional memory during HS recovery [44,45] (Figure 2). In recent years, a lot of effort has been made to elucidate the role of histone acetylation in mediating heat stress-induced responses. More specifically, under high temperatures *Arabidopsis HISTONE DEACETYLASE 3 (HD2C)* transcript increase leads to the suppression of heat-inducible genes, a molecular event that is facilitated through the physical interaction of HD2C with the chromatin remodeler BRM [46]. An additional interactor of HD2C, HDA6 (HISTONE DEACETYLASE 6), promotes thermotolerance through the RdDM pathway as well as independently of the latter [47,48] (Figure 2). In particular, *hda6* mutants demonstrate hypersensitivity and a significantly disorganized transcriptome during heat stress treatment [48]. Furthermore, the *Arabidopsis* histone chaperones ANTI- SILENCING FUNCTION 1A and 1B (ASF1A and ASF1B) mediate the transcriptional activation of the heat-responsive genes *HSFA2* and *HSA32* through nucleosome dissociation and H3K56 acetylation of these loci [49] (Figure 2). GCN5 (GENERAL CONTROL NONDEREPRESSIBLE 5) is a HAT that also activates the expression of the thermotolerance-conferring *Arabidopsis* genes *HSFA3* and *UVH6* (*ULTRA-VIOLET HYPERSENSITIVE 6*) by promoting H3K9 and H3K14 acetylation at their promoter regions [50] (Figure 2).

#### 2.2.4. DNA Methylation

As a pivotal epigenetic mark, DNA methylation is involved in gene regulation during plant HS responses [45,51]. It has been shown that HS results in a global increase in the DNA methylation levels in *Arabidopsis* [52]. A global increase in the level of DNA methylation, is in turn linked to the function of the RdDM pathway, which contributes to the acquisition of thermotolerance in plants [48]. In particular, HS triggers the upregulation of components involved in the RdDM, such as the DNA methyltransferase DOMAINS REARRANGED METHYLTRANSFERASE 2 (DRM2) and the RNA polymerase IV (PolIV) and PolV subunits, NUCLEAR RNA POLYMERASE D1 (NRPD1) and NUCLEAR RNA POLYMERASE E1 (NRPE1) [53] (Figure 2). 

HS tends to activate certain TEs which enhance plant thermotolerance [51,54]. The HS-induced temporary activation of the TE ONSEN helps strengthen heat-tolerance [55,56], while DNA methylation represses ONSEN activation together with histone H1 and other factors in mature plants [51] (Figure 2). Overall DNA methylation, in response to heat stress, acts as a ‘buffer’, helping maintain a balance between the heat stimulus and the acute activation of TEs [51].

Epigenetic changes also take place during the recovery period from HS, to facilitate a plant to develop a HS memory. Differentially methylated cytosine (DMC) analysis in *Arabidopsis* under HS, HS recovery, and control temperature conditions, depicted that DNA methylation increased during HS and decreased during the recovery period [57]. Additionally, histone H3K4 methylation negatively affects the RdDM pathway and results in hypomethylation [58]. Recently, two Trithorax (TrxG) protein family members, histone H3K4 methyltransferases ARABIDOPSIS TRITHORAX-RELATED 7 (ATXR7) and ARABIDOPSIS TRITHORAX 1 (ATX1) which are significantly upregulated in response to heat stress, were demonstrated to be implicated in the downregulation of histone and DNA methylation at certain HS-associated gene loci during both heat stress and recovery [45] (Figure 2). The activation of HS genes and heritable TEs contributes to the establishment and maintenance of plant HS memory, helping improve the plant thermotolerance in future HS conditions [45].

#### 2.2.5. Non-Coding RNA (ncRNA) Regulation of Heat Stress Responses

Non-coding RNAs, including microRNAs (miRNAs), small interfering RNAs (siRNAs), circular RNAs (circRNAs) and long non-coding RNAs (lncRNAs), are actively involved in the plant HS response and memory [59]. Plant miRNAs repress the expression of their targeted genes [60]. In particular, miR398 contributes to the down-regulation of several HS-related genes such as *COPPER/ZINC SUPEROXIDE DISMUTASE 1/2* (*CSD1/2*) and *COPPER CHAPERONE FOR SUPEROXIDE DISMUTASE* (*CCS*) [61]. *SQUAMOSA PROMOTER BINDING PROTEIN-LIKE* (*SPL*), target of miR156, is downregulated under HS, and further contributes to the expression of flowering time promoters *FLOWERINGLOCUS T* (*FT*) and *FRUITFULL* (*FUL*) [60]. The *GIBBERELLIC ACID MYB* (*GAMYB*) transcription factor loci are targets of miR159 and these transcription factors contribute to the improvement of the plant HS tolerance [61]. Plant siRNAs are involved in the RdDM-based *de novo* methylation [62]. 

The siRNA biogenesis is involved in the regulation of ONSEN. Plants deficient in siRNAs exhibit constitutive HS memory phenotypes and the activation of ONSEN is heritable in this case [63]. Another category of ncRNAs, circRNAs, are involved in transcriptional or post-transcriptional regulation, and are significantly upregulated under HS [64]. More than 1500 heat-specific circRNAs have been identified so far [64,65]. These circRNAs potentially regulate plant heat responses by acting as miRNA-binding competitors [65]. The former research identified three miR9748-targeted hormone signaling-related genes (Csa1M690240.1, Csa7M405830.1, Csa6M091930.1) which were upregulated by competitive interactions among miR9748 and two newly-found circRNAs (novel_circ_001543, novel_circ_000876) [63]. MiR9748 has been shown to have an effect on HEAT-SHOCK PROTEIN 90 (HSP90) that is involved in protein processing at the endoplasmic reticulum [65]. 

## 3. Conclusions and Future Perspectives

Increases in environmental temperature have a great effect on plant growth and development. Considering how important agriculture and crop production sustainability is for the global population, better understanding and deciphering the mechanisms plants use to perceive and respond to different levels of warmer temperature, is a priority. Warm temperature responses of plants can be categorized as either thermomorphogenic or heat stress responses, depending on the temperature levels. 

Epigenetic mechanisms such as chromatin regulation and remodeling are essential for shaping warmer average temperature responses [2,27]. Therefore, it is essential that additional studies are performed to contribute to further analysis and mapping of the epigenetic mechanisms involved in warm temperature responses. Established techniques such as Chromatin Immunoprecipitation as well as novel technologies such as Chromatin Conformation Capture (Hi-C) performed on plants grown under specific temperature conditions, followed by parallel next generation sequencing of lines overexpressing or lacking components involved in temperature perception and signaling could provide insight on the regulation of global chromatin remodeling levels and possibly identify new interactions. Furthermore, chromatin remodeling complexes can be further investigated through studies involving mass spectrometry analysis, performed on plants exposed to specific temperature conditions. Additionally, further investigation focusing on potential overlapping of specific epigenetic mechanisms involved in heat stress response and warm temperature thermomorphogenesis (ex. SUMOylation) would be very informative. The aforementioned experiments could provide even further insights into the epigenetic mechanisms involved in warmer temperature integration and signaling responses in plants, which could subsequently offer means of sustaining global crop yield and productivity.

Plant adaptation to heat stress also relies on a number of epigenetic mechanisms, including chromatin remodeling, chromatin-associated SUMOylation [26], histone modifications [40,44,45,46,47,48,49,50] and ncRNA regulation [60]. Although a lot of effort has been put on elucidating the epigenetic adaptation to heat acclimation, the current literature is still lacking evidence on how different factors such as transcription factors, ncRNAs and chromatin modifications cooperate to configure heat stress responses. Another interesting topic that has yet to be fully investigated, is the role of histone modifications in the acquisition of transcriptional memory to heat stress. Furthermore, the contribution of chromatin remodeling complexes to plant thermotolerance is quite limited and further research is required to identify additional molecular complexes and factors that could potentially be involved.

In the last decade the global interest in understanding plant heat stress responses has increased, due to the unprecedented effects of global warming and overpopulation. The rise of average global temperatures poses a detrimental threat to global crop productivity affecting nutrient absorption and availability in plants, as well as impacting normal growth and physiological responses. Hence, better understanding and uncovering how warm temperature alters chromatin dynamics and nuclear organization, will contribute to the efforts to develop precision agriculture practices and promote global food security. Developing molecular and epigenetic strategies to improve crop resilience to increased temperatures would provide breeders the tools to maximize production in response to climate change. For instance, utilizing current and future knowledge of epigenetic marks could contribute to the discovery of new epialleles and/or enrichment of phenotypic variation. In conclusion, epi-breeding is a valuable tool which can be used to manipulate plant transcriptomes to enhance thermotolerance and priming that could prevent and manage crop losses.

## Figures and Tables

**Figure 1 plants-10-02439-f001:**
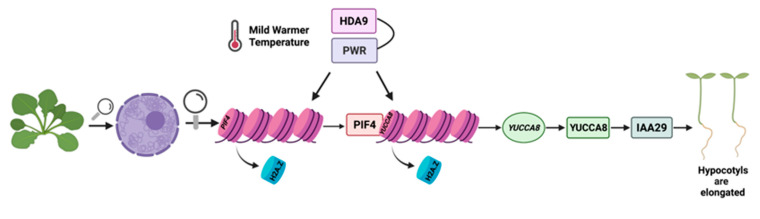
The epigenetic events underlying warmer average temperature responses. Under mild warmer average temperatures hypocotyl elongation is induced through the action of different factors. POWERDRESS (PWR) and HISTONE DEACETYLASE 9 (HDA9) physically interact, resulting in histone deacetylation of specific loci. This event is important for the subsequent exclusion of H2A.Z from +1 nucleosomes of *PIF4* and *YUCCA8*. PWR acts upstream of the hypocotyl elongation factor PIF4 as a chromatin remodeler facilitating this process. The above events result in the ability of PIF4 to bind the *YUCCA8* promoter. YUCCA8 protein synthesis in turn leads to the expression of further auxin-regulating genes such as *IAA29*, which further leads to IAA29 protein synthesis and subsequent hypocotyl elongation. Proteins are depicted in squares and genes in circles. The figure was created in BioRender.

**Figure 2 plants-10-02439-f002:**
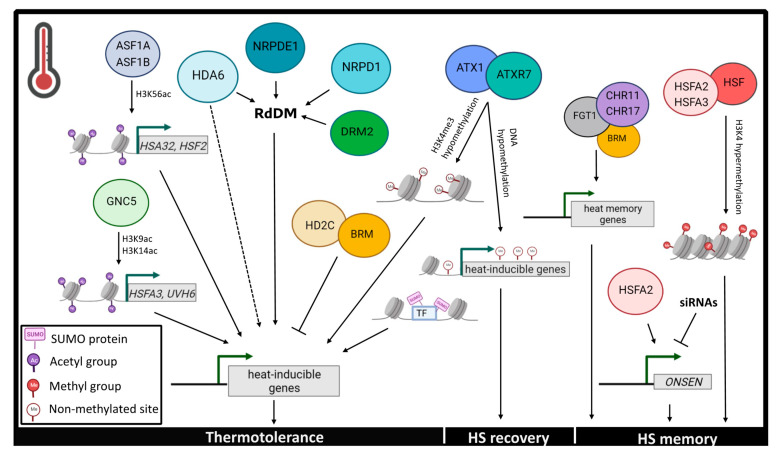
Epigenetic events controlling temperature responses to heat stress. Plants utilize epigenetic mechanisms such as histone modifications, chromatin remodeling and DNA methylation to cope with heat stress and acquire heat stress memory. Epigenetic marks contributing to thermotolerance include H3K9, H3K14 and H3K56 acetylation, H3K4 hypomethylation, *de novo* DNA methylation mediated through the RdDM pathway and chromatin SUMOylation. DNA hypomethylation of heat inducible genes is an essential step in HS recovery, while chromatin remodeling, HSFs and H3K4 hypermethylation contribute to the acquisition of HS memory. Dotted lines symbolize unknown factors involved in the pathway illustrated. Heat Stress (HS), Heat Shock Factors (HSF), RNA-directed DNA Methylation (RdDM). The figure was created in BioRender.

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
