# Peer review of "The Epigenetic Mechanisms Underlying Thermomorphogenesis and Heat Stress Responses in Arabidopsis"

_plants, 2021, doi:10.3390/plants10112439_

Round 1
Reviewer 1 Report
Dear authors,
A very nice piece of work, one which I enjoyed reading, well done.
I have identified a number of concerns / issues however throughout the manuscript which I would like to see addressed as I believe this would improve your manuscript further.
My issues / concerns / comments are identified on the attached annotated PDF which has been uploaded for your reference.
Please also use the MDPI supplied MS Word template when putting together your revised manuscript.
For example;
The Figure headings and legends are different to how you have them currently formatted - please address this using the template.
Figure placement also needs to be determined by the authors.
The referencing format in text throughout the manuscript is incorrect - please address using the template.
The reference list itself is also in the incorrect format - please address using the template provided by MDPI Plants.

Author Response
Thank you very much for your very helpful and constructive comments. We believe that we have now addressed all points raised as follows and highlighted in the revised text file.
In the introduction more detail has been added to provide a stronger link between Arabidopsis thaliana research and its subsequent translation to crop species.
The spelling has been checked to always comply with UK English. In all parts where the general level of English needed to be improved, sentences have been made clearer and more concise and choices of certain words have been replaced.
The Nuclear re-organization section (2.2.2) has been expanded to provide more information.
Figure 2 caption has been updated to be shorter and more generally descriptive of the mechanisms depicted there.
The position of Figures has also changed, they have been inserted in the main text close to where their citation is in the text, and their format has been updated to comply with the MDPI one. There are now clear indications in the text for where the Figures should be placed, and the images of the Figures are provided in a separated zip file as indicated by the journal instructions.
Referencing has also been updated to comply with MDPI format.
Reviewer 2 Report
This review gives a focussed state-of-the-art overview on epigenetic processes underlying response to warm temperature/heat. It will be of great interest for researchers in both fields, thermotolerance and heat stress responses of plants, but also for people not directly in this field, but rather interested in general plant stress responses. Therefore, I think it is highly topical.
In general, the review is a kind of short review, giving concentrated information and not a broad, detailed overview. This makes it easier to read for researchers and helps to just extract first clues in the field. It s well and understandable written. I have only some minor comments:
- Figure 1, Legend: Should read POWERDRESS
- Figure 1, Legend: The sentence
"This event is key for the subsequent exclusion of H2A.Z from +1 nucleosomes of PIF4 and YUCCA8" is, when looking at the Figure, misleading. As shown in the Figure, PIF4 is somehow upstream from YUCCA8. And in addition HDA9 and PWR exclude from both genes (PIF4 and YUCCA8) H2A.Z. It is more complex than the sentence states. This should be explained in more detail.
- In conclusion and future perspectives, the conclusions are nicely elaborated. However, future perspectives could be addressed in more detail (how exactly the knowledge can be utilized to get better crops.....).
Author Response
Thank you very much for your very helpful and constructive comments. We believe that we have now addressed all points raised as follows and highlighted in the revised text file:
Figure 1 and its caption have been corrected and adjusted in order to be clearer in explaining the mechanism of action of the genes and proteins mentioned in the figure, especially PIF4 and YUCCA8.
With regards to the Conclusion and Future Perspectives, a new paragraph had been added addressing into more depth and detail how the knowledge on epigenetic mechanisms in warm temperature and heat stress response will help provide practical methods to be used in crop improvement.